# Avian Influenza: Could the H5N1 Virus Be a Potential Next Threat?

Elena Imperia [1,2], Liliana Bazzani [3], Fabio Scarpa [4], Alessandra Borsetti [5], Nicola Petrosillo [6], Marta Giovanetti [3,7,*] and Massimo Ciccozzi [1,*]

1    Unit of Medical Statistics and Molecular Epidemiology, University Campus Bio-Medico of Rome, 00128 Rome, Italy
2    Unit of Gastroenterology, Department of Medicine, University Campus Bio-Medico of Rome, 00128 Rome, Italy
3    Sciences and Technologies for Sustainable Development and One Health, University of Campus Bio-Medico of Rome, 00128 Rome, Italy
4    Department of Biomedical Sciences, University of Sassari, 07100 Sassari, Italy
5    National HIV/AIDS Research Center (CNAIDS), National Institute of Health, 00161 Rome, Italy
6    Infection Prevention and Control—Infectious Disease Service, Fondazione Policlinico Universitario Campus Bio-Medico, 00128 Rome, Italy
7    Rene Rachou, Fundação Oswaldo Cruz, Belo Horizonte 21040-360, Brazil
*    Correspondence: giovanetti.marta@gmail.com (M.G.); m.ciccozzi@unicampus.it (M.C.)

**Abstract:** Avian influenza virus (AIV) poses a significant challenge to poultry production, with negative repercussions for both the economy and public health worldwide. Since January 2003, a total of 868 human cases of AIV H5N1 have been reported from four countries in the Western Pacific Region, as of 9 March 2023. When AIVs are circulating in poultry, there is a risk of sporadic infections and small clusters of human cases due to exposure to infected poultry or contaminated environments. The increase in reported A(H5N1) infections may reflect continued virus circulation in birds, as well as enhanced surveillance and diagnostic capacity resulting from the response to the COVID-19 pandemic. Numerous countermeasures, including vaccines and antiviral treatments, are available for influenza infection. However, their effectiveness is often debated due to the ongoing resistance to antivirals and the relatively low and unpredictable efficiency of influenza vaccines compared to other vaccines. Vaccination remains the primary method for preventing influenza acquisition or avoiding serious complications related to the disease. In this review, we summarize the global landscape of the Influenza A virus and provide insights into human clinical symptomatology. We call for urgent investment in genomic surveillance strategies to timely detect and shape the emergence of any potential viral pathogen, which is essential for epidemic/pandemic preparedness.

**Keywords:** avian influenza virus (AIV); H5N1; Hemoagglutinin (HA); Neuramidase (NA); epidemic/pandemic preparedness

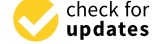



## 1. Introduction

Avian influenza virus (AIV), a member of the *Orthomyxoviridae* family, remains a challenge for poultry production, with negative repercussions for both the micro- and the macro-economy and public health around the world. Globally, from January 2003 to 26 January 2023, there were 868 cases of human infection with avian influenza A(H5N1) virus reported from 21 countries. Of these 868 cases, 457 were fatal [1]. Currently, three types of influenza viruses, based on their antigenic differences, have been described: A, B, and C. Types A and B cause the annual influenza epidemics, which have up to 20% of the population sniffling, aching, coughing, and running high fevers. Type C also causes flu, but its symptoms are much less severe. In addition to these three types, influenza D virus was first identified in pigs and then found to be prevalent in livestock [2–5].

Human pathogens predominantly bind to glycosylated proteins characterized by a terminal sialic acid (SA) α-2,6-galactose (Gal) residue, while avian influenza microorganisms, including H5N1 virus, preferentially bind to SA α-2,3-Gal [6]. AIV primarily infects the respiratory tract, and the main clinical manifestations in humans include nasal discharge, high fever, and weight loss due to dehydration. Severe pathogenesis is represented by lung damage due to extensive infiltration of the lung tissue by inflammatory cells [7].

In response to increasing awareness of circulating AIV strains, the World Health Organization (WHO) closely examines viral antigenicity through sentinel laboratories worldwide to select strains for the development of candidate vaccines for pandemic/epidemic preparedness. To date, vaccines are available for H5, and several drugs, such as Oseltamir, Peramivir, and Zanamivir, have been produced and administered to prevent severe cases [8].

AIV cases may increase globally through migrating birds and poultry trade transmission routes [9]. Poultry vaccination is now considered a relevant and effective control measure. However, the accumulation of point mutations in the viral genome may negatively impact the efficacy of poultry vaccination until a correctly matched vaccine is selected, manufactured, and administered in a timely manner [10]. For this reason, active monitoring of circulating strains is required to constantly prevent the likely emergence of novel viral mutants with epidemic/pandemic potential in both the poultry and human populations [10]. In the last century, the most severe pandemic was the "Spanish Influenza", caused by the H1N1 virus, as described by Sutton in 2018 [11,12]. After the 1918 pandemic, H1N1 viruses continued to circulate in humans, and these viruses displayed reduced morbidity and mortality. However, viral reassortment between the 1918 H1N1 strain and AVI viruses resulted in two more pandemics related to the Asian Influenza H2N2 and the Hong Kong Influenza caused by H3N2, which were recorded in 1957 and 1968, respectively [9,11,13]. More recently, in the spring of 2009, a new strain of the influenza A virus (H1N1), initially detected in the United States [14], rapidly spread across the world [15]. This novel strain of H1N1 contained unknown combinations of influenza genes not previously identified in animals or humans and was therefore renamed the influenza A (H1N1) pdm09 virus [16,17]. The strain was markedly distinct from the H1N1 strains known up to that point, and as a consequence, only a small fraction of young people had any pre-existing immunity [18]. Due to the antigenic differences between the (H1N1) pdm09 virus and the circulating H1N1 viruses, the seasonal flu vaccines provided limited protection against this pandemic virus. From 12 April 2009 to 10 April 2010, the CDC estimated 60.8 million cases of H1N1 influenza in the United States alone [19].

In this review, we summarize the global landscape of H5N1 and its evolution to provide evidence regarding the impact in the implementation of a global genomic surveillance system to enable early detection and characterization of novel variants that would direct the global epidemic response.

## 2. Avian Influenza Virus: Genomic Epidemiology

The influenza virus genome is composed of 13 kb and encodes 12 proteins (Figure 1): hemagglutinin (HA), neuraminidase (NA), M1 matrix protein (M1), M2 ion channel protein (M2), nucleocapsid protein (NP), nonstructural protein (NS1, NS2), and RNA polymerase complex (PB1, PB2, PA, PB1-F2 and PA-X) [20]. Influenza A viruses are subtyped based on their combination of HA and NA surface glycoproteins, and up to know a total number of 18 HA and 11 NA subtypes have been already described [21–23]. HA is a glycoprotein formed by three district regions. To date, more than 16 different subtypes of HA have been identified and recently classified in two distinct groups and four different clades: (i) group 1, which contains the H1 clade (H1, H2, H5, H6, H11, H13, H16) and the H9 clade (H8, H9, H12); and (ii) group 2, which includes the H3 clade (H3, H4, H14) and the H7 clade (H7, H10, H15) [20,24].

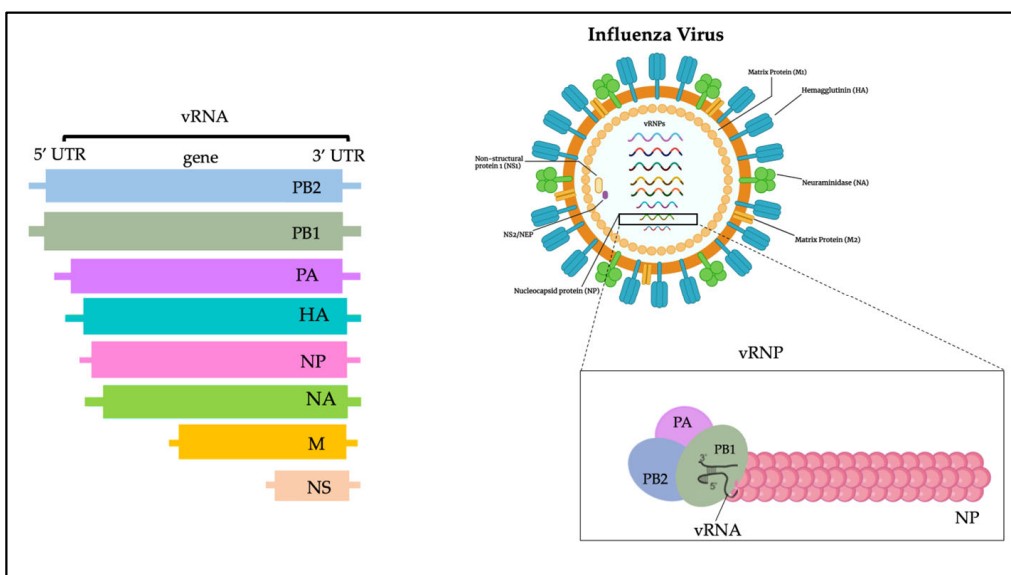

**Figure 1.** Influenza virus genome.

NA is instead a tetramer of four identical monomers known as the catalytic head, the stalk, the transmembrane region, and the cytoplasmic tail. NA subtypes are divided into three distinct groups: group 1 (N1, N4, N5, N8); group 2 (N2, N3, N6, N7, N9); and group 3, which includes NA influenza B viruses. N10 and N11 subtypes have been found only in bats [24].

The HA and NA are the prime determinants of the pathogenicity of influenza A viruses. HA attaches virions to cells by binding to terminal sialic acid residues on glycoproteins/glycolipids to initiate the infectious cycle, while NA cleaves terminal sialic acids, releasing virions to complete the infectious cycle [20]. Antibodies specific to HA or NA can protect experimental animals from AIV pathogenesis and drive antigenic variation in their target epitopes that impairs vaccine effectiveness in humans [9,20].

M1, M2, NS1, and NS2 proteins enhance cytoplasmic trafficking of vesicles, are critical for cytoplasmic trafficking of viral ribonucleoproteins (vRNPs), and aid in the assembly of AIV virions [25]. The M1 protein has a molecular weight of about 26 kDa, and during the last stages of viral replication it translocates into the nucleus and helps to inhibit viral transcription. Subsequently, M1 interacts with the NS2 tail encoded by the viral RNA fragment to expel the vRNPs from the nucleus to the cytoplasm, thus triggering the budding process and the release of virions [26]. The M2 protein is a type III tetrameric integral transmembrane protein and plays an essential role in viral replication by mediating the acidification and shedding of the endosomal trapped virus envelope. This protein is a proton channel activated by low pH; after endocytosis and before HA-mediated fusion between viral and endosomal membrane, M2 channels are activated by low pH of the endosome to conduct protons to acidify the viral interior [27]. Such acidification weakens the electrostatic interaction between ribonucleoprotein (RNP) complexes and matrix proteins. Consequently, membrane fusion can release uncoated RNPs into the cytosol for transport to the nucleus [27].

In the cell replication cycle, NS1 is involved too; it is situated into the nucleus, and its nuclear functions include inhibition of host mRNA processing, blocking of host mRNA nuclear export machinery, and the general inhibition of the host antiviral [28].

NS2 is important in the virus life cycle; it rules the transcription and replication of viral RNA: it is implicated in the regulation of the accumulation of genomic vRNA, antigenomic cRNA and mRNA synthesized by the viral RNA-dependent RNA polymerase. Furthermore, NS2 has been shown to be an important factor in the adaptation of highly pathogenic H5N1 avian influenza viruses to the mammalian host [29].

### 3. H5N1 Epidemiologic Insights

In 1997, H5N1 was initially discovered in a three-year-old boy from China, who suffered from acute pneumonia and respiratory distress syndrome. Following the first diagnosis, an additional seventeen patients were identified, of which six fatalities were reported [30]. Human cases were preceded by outbreaks in poultry, and in December 1997, the epidemic in Hong Kong was contained by the slaughter of all poultry in markets and farms [31].

In late 2003 and early 2004, an H5N1 outbreak occurred in poultry farms across several countries, including Cambodia, China, Indonesia, Japan, Laos, South Korea, Thailand, and Vietnam, resulting in two deaths among two confirmed and one probable case [21,31–33]. The virus subsequently spread to ten other Asian countries in 2004 due to the poultry trade. In Thailand, 17 cases and 12 deaths were reported, while in Vietnam, there were 28 cases and 20 deaths [34]. The virus then spread further afield, reaching Central Asia, South Asia, the Middle East, and parts of Africa in 2005 via migratory birds [35].

As of November 2003, a total of 861 human cases of H5N1 infection and 455 deaths have been reported from 17 countries, with a cumulative mortality rate of more than 50% [4].

The first H5N1 case in Egypt was confirmed in March 2006, with the Egyptian Ministry of Health reporting 6355 suspected cases of H5N1 infection and 112 confirmed cases resulting in 36 deaths between March 2006 and March 2009, according to a WHO report. Of the confirmed cases, all except for two were linked to infected poultry [36]. On 10 August 2012, 608 people in 15 countries were infected with H5N1, with over 100 cases reported. Indonesia, Vietnam, and Egypt had the highest number of human cases [37].

Multiple H5N1 infections in domestic poultry and wild birds have occurred in more than 60 countries since September 2017, with sporadic human infections occurring in 16 countries [38]. The Italian National Reference Center (CRN) for avian influenza and Newcastle disease (based in Rome, Italy) confirmed the first positivity for highly pathogenic avian influenza virus (HPAIV) subtype H5N1 in poultry on 22 September 2022. The Delegated Regulation 2020/687 required the implementation of general and specific control measures in the protection and surveillance zones. When available, the duration of these measures is indicated in the document "Outbreaks in Italy". The CRN confirmed the first HPAIV virus subtype H5N1 positivity in wild birds on 29 September 2022. See the document "Positivity in Wild Animals in Italy" [39] for more information on the species involved.

On 23 February 2023, Cambodia's national focal point for international health regulation notified the WHO regarding a confirmed case of H5N1 virus infection in humans. An 11-year-old girl from Prey Veng province in southern Cambodia developed symptoms on 16 February 2023, and was treated at a local hospital. On 21 February 2023, the case was admitted to the National Pediatric Hospital with severe pneumonia, and a sample was taken as part of the activities of the severe acute respiratory infection sentinel surveillance system (SARI-). On the same day, the sample tested positive for avian influenza A virus (H5N1) using reverse polymerase chain reaction (RT-PCR) at the National Institute of Public Health. The sample was then sent to the National Center for Influenza at the Institute Pasteur Cambodia, where on 22 February 2023 [40], it was confirmed that the patient had died.

### 4. Mechanisms of H5N1 Virus Transmission to Humans and Associated Molecular Signature

AIV, based on the pathobiological effect in chickens, are classified as low pathogenic (LPAIV) or highly pathogenic (HPAIV) strains.

Commonly, LPAIV strains cause mild and asymptomatic infections in wild aquatic birds, but when introduced into domesticated poultry, infections may be asymptomatic or produce clinical manifestations and lesions showing pathophysiological damage to the respiratory tract, gastrointestinal and re-productive systems.

The HPAIV strains have primarily been isolated in gallinaceous poultry, where they are mainly linked with a high morbidity and mortality. Although HPAIV have rarely infected

domestic waterfowl or wild birds, the Eurasian-African H5N1 HPAIV have evolved over the past decade with the unique capacity to infect and cause disease in domestic ducks and wild birds [41–43].

HPAIV virulent strains can induce severe disease and cause devastating outbreaks with high mortality rates in poultry [44]. LPAI infections tend to localize in the mucosal surfaces of the gastrointestinal tract of infected birds, and although often asymptomatic, chickens may present with mild clinical signs following infection [45].

In general, direct transmission of AIV viruses to human from poultry probably occurs in intensive farms when the operator does not use the appropriate devices [9]. H5N1 have a binding preference for sialic acid residues with an $\alpha$-2,3-Gal terminating sequence, which is present on the surface of avian receptors [46], so AI viruses need to modify the HA binding site to interact with humans, which present an $\alpha$-2,6-Gal terminating sequence [9].

The host cells are made up of complexes, including polysaccharides (N-linked glycans, O-linked glycans, glycolipids, etc.), necessary for the cells to carry out their vital activities and through which the influenza viruses are attacked. H5N1 binds its own HA protein to sialic acid receptors expressed on the surface of epithelial cells in the host's enteric and respiratory tracts. Sialic acid is a monosaccharide made up of nine carbon atoms. The two most common forms of salic acids are: N-acetylneuraminic acid (Neu5Ac) and N-glycolylneuraminic acid (Neu5Gc). The former is regulated by the enzyme CMP-Neu5Ac hydroxylase (CAMH) to be hydroxylated and form Neu5Gc [47]. Influenza viruses from different host sources have different binding abilities to these two forms of sialic acid. However, depending on how the terminal sialic acids bind to the sub-terminal galactose of the chain, they can be more or less related to the influenza virus, in fact, the carbon atom in position C2 of the sialic acids can be connected to the carbogen at the C3 or C6 position of the secondary terminal galactose via an $\alpha$-2,3 sialyltransferase (ST3Gal) or $\alpha$-2,6 sialyltransferase (ST6Gal) catalyzed by $\alpha$-2,3 or $\alpha$-2,6 glycosidic linkage, and the effect of different sialic acids of $\alpha$-2,3 or $\alpha$-2,6 glycosidic linkages on the binding properties of influenza virus receptors is currently commonly regarded as an important factor limiting the interspecies transmission of influenza viruses [47].

Belser et al., in 2011 and 2014, performed two studies on animal models, with the aim of assessing airborne transmission of the virus in humans and mammalian, and concluded that ferrets were the preferred model because are very similar to humans: they presented clinical signs upon influenza infection [48], had similar receptors on cell surface (i.e., sialic acid) [49,50], and efficiently transmitted (i.e., transmission to 75–100% of contacts) pandemic influenza viruses via direct contact and respiratory droplets [21] (Figure 2B,C). The most common amino acid substitution for LPAI strains to increases avian-to-mammalian receptor affinity is an HA Q226L substitution [45,51–53], but Q226L mutation has not been commonly observed in the HA of H5N1 HPAIV [54]. In any case, HA Q192H, HA I151T and HA Q222L are alternative substitutions, especially the latter, which confers increased ability for replication and transmission in human hosts and ferret model [2,54]. Moreover, PB2 E627K change in H5N1 HPAI has also been shown to be crucial to allow virus transmission in ferrets [48,55]. Belser et al. demonstrated, using an animal model, that introducing the mutations HA Q222L and G224S (Q226L and G228S, H3 numbering), and PB2 E627K, followed by 10 serial passages, with the acquisition of additional mutations PB1 H99Y, HA H103Y and T156A (H110Y and T160A, H3 numbering) results in airborne transmission between ferrets [21,48,56]. Among these mutations, HA H103Y and T156A were found to reduce the pH of fusion and increase the temperature stability of the virus and enhance binding to both $\alpha$-2,3 and $\alpha$-2,6-SAs, respectively. Another crucial point in increasing H5N1 viral replication, polymerase activity and transmission is the presence of a multi-basic cleavage site (MBCS), so the virus can infect a great number of cells and lead to systemic infection which [57,58]. In fact, the presence of an MBCS in the HA of influenza A viruses allows HA cleavage by other enzymes such as furin-like proteases [59].

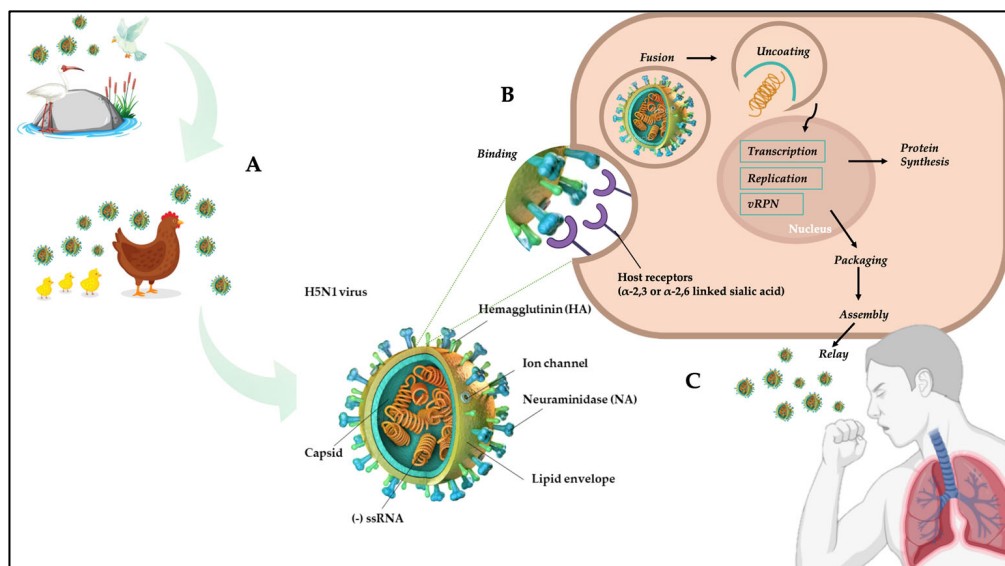

**Figure 2.** H5N1 transmission cycle. (**A**) Transmission of AVI from aquatic birds to domestic poultry. (**B**) Pathogenic mechanism of virus, local replication. (**C**) Clinical manifestation and diffusion to the environment.

## 5. Symptomatology, Laboratory Diagnosis and Treatment

The main route of transmission for H5N1 infection in humans is bird-to-human transmission. In fact, infected patients reported a history of recent exposure to dead or ill poultry [60]. Current data indicate that the incubation period might last 7 days or less, with a mean of 2 to 5 days [54]. As reported by Metha et al., infected humans present mainly respiratory symptoms but also gastrointestinal or central nervous system features [60,61]. Occasionally, respiratory symptoms may be accompanied by headache, myalgia, sore throat, rhinorrhea, or uncommonly conjunctivitis or bleeding gums. Severe cases might present multiorgan failure, including renal dysfunction, pulmonary hemorrhage, pneumothorax, and pancytopenia. Respiratory failure can lead to death, and this might be enhanced by diffuse alveolar damage, hyaline-membrane formation, lymphocytic infiltration into the interstitial region, and the presence of reactive fibroblasts [60,62]. Laboratory abnormalities generally include transaminitis (AST > ALT), elevated levels of lactate dehydrogenase, creatine kinase, and hypoalbuminemia [63]. Vaccines and drugs are key points in reducing virulence and the likely accumulation of mutations. HA and NA are the main drug targets for the influenza virus since they are essential for viral replication. Amantadine and rimantadine are among the main antiviral drugs currently available. They interfere with the release of infectious viral nucleic acid into the host cell through interaction with the transmembrane domain of the M2 protein of the virus. They also appear to prevent virus assembly during replication in some cases. Recently released cases showed an increasing incidence of resistance against this treatment, and for this reason, a global effort is trying to search for drugs (including la-ninamivir octanoate, zanamivir, and peramivir) that might be able to target both the NA and the M2 viral proteins, with a primary focus on NA inhibition [64,65].

Despite the use of antiviral medications, vaccination is critical for influenza control and prevention. The glycoproteins haemagglutinin and neuraminidase are directly associated with infection protection and illness amelioration. There are three types of inactivated influenza trivalent vaccine available worldwide: whole virus, split product, and purified surface antigen [66]. Each type of vaccine has benefits and drawbacks. Vaccines can activate immune effectors such as humoral effectors (antibodies produced by B lymphocytes) that can bind specifically to virus antigens, and cellular effectors (cytotoxic CD8+ T lymphocytes) that can limit virus spread inside the body by killing infected cells. All influenza vaccines must be updated annually because immunity is only temporary [67],

and to match the antigenicity of the circulating viruses [68]. As a result, the WHO and other stakeholders meet biannually to select the most appropriate influenza viruses from globally circulating viruses based on antigenic and genetic characteristics, as well as epidemiological data from various countries. [69]. Trivalent vaccines containing two influenza A viruses (H1N1 and H3N2) and one lineage of influenza B virus were formulated to provide protection against three distinct influenza viruses for many years, despite the existence of two different lineages of circulating B viruses. To provide extensive exposure against the current circulating viruses, a second lineage of B virus was added to the formulation for a quadrivalent influenza vaccine. Current approved influenza vaccines are quantitatively standardized in terms of HA quantity or antigenicity, but not in terms of neutralizing antibodies (NA). As a result, both the NA content and the NA immunogenicity of vaccinations may vary [70]. The inactivated virus vaccine (IIV) is the most popular worldwide approach due to its high safety and relatively low production costs. Consequently, it has the highest percentage in the flu vaccine global market [70]. In general, the virus is produced in embryonated chicken eggs or cultured cells of mammalian origin for this type of vaccine. It has been demonstrated that IIV can induce local and systemic immunity, although booster vaccinations might be needed to maintain the antibody titers. [70]. The inactivated influenza vaccine can be categorized into three types: whole-virus inactivated vaccines, split-virus inactivated vaccines, and subunit inactivated vaccines [71]. The first vaccine to be introduced to the market was the whole-virus vaccine (WIV) type. However, it was associated with systemic and local side effects, which could result from impurities such as egg proteins [72]. Modern vaccine production technologies have resulted in safer vaccines with fewer side effects and impurities. WIV virions are chemically inactivated with formaldehyde or β-propiolactone after incubation and growth in embryonated eggs, and then purified. Although this method is still commonly used, other methods such as heat and radiation can also be used [73]. Split-virus and subunit vaccines are more commonly used due to their reasonable immunogenicity and ease of production, despite losing some of the inherent immunogenicity by not including the whole virus structure [74]. Split-virus inactivated vaccines aim to induce a neutralizing antibody immune response [75]. These vaccines are subtype- and often strain-specific, requiring accurate prediction of circulating viral strains during an outbreak [76]. The vaccine is prepared by chemically disrupting the virus membrane using surfactants, followed by removal of the surfactant by tangential flow filtration. Virus-inactivated split vaccines induce the production of antibodies against the globular head of the hemagglutinin [76]. Subunit vaccines only contain antigenic parts of the virus, HA and NA, which are purified from the influenza virus after splitting the virus using surfactants. This type of vaccine usually requires an adjuvant to augment its effectiveness, particularly in the senior population [70]. In past clinical trials, recombinant HA (H5N1) subunit vaccines at a 90 μg HA dose administered with a prime-boost schedule induced only modest responses [77]. Currently, the leading H5N1 vaccine adjuvants are oil-in-water (o/w) emulsions that increase neutralizing antibody titers, implement the breadth of cross-reactive antibodies, and possess dose-sparing activity as well [77]. The first novel adjuvant approved in 2015, applied in seasonal influenza vaccines (e.g., FLUAD) is MF59, a squalene-based oil-in-water emulsion [78]. The first proposal for live attenuated vaccines (LAIV) was in the 1960s by growing the influenza viruses under suboptimal conditions in eggs [79]. The live attenuated influenza vaccines were produced to mimic natural infection and immunity without causing severe illness and consequently inducing humoral and cellular immunity [79]. These attenuated viruses are temperature-sensitive and grow only at 25 °C (cold-adapted). Cold-adapted donor viruses receive several passages with a gradual reduction in temperature in embryonated chicken eggs. Thus, LAIV can grow at the same temperature range as the nasopharynx's mucosal surface when administered intranasally [80]. LAIV presents some advantages than IIV as it mostly provokes local mucosal immunity and local immunoglobulin A (IgA) production as well as IgG [80], in contrast to systemic IgA and IgG antibodies only with IVV. However, LAIV is not recommended for immunocompromised individuals or people who are in close contact with

them due to the use of live viruses for immunization [80]. Recombinant HA vaccines can be produced using recombinant protein expression technology by insect cells and baculovirus because of their high yields and cost-effectiveness [81]. The stereotypical cell lines used for recombinant protein expression are insect cells, and the Gibsco Sf9 cell line is the most popularly applied in human or veterinary medicine for recombinant protein expression [81]. The main advantage of recombinant HA vaccines is the absence of unwanted mutations from egg adaptations, making them suitable for people who suffer from egg allergy [82]. Recombinant HA vaccines have a similar mechanism of action to IIV, but they are less immunogenic and require three times more HA than IIV to elicit the same antibody titers [82]. Another positive aspect is that recombinant HA vaccines are appropriate for an influenza pandemic due to the absence of highly pathogenic viruses, making them safer than other influenza vaccines [83]. However, these vaccines are less effective in children, and their use is limited to adults [83].

## 6. Conclusions

Influenza remains a major public health concern because it causes annual epidemics and has the potential to trigger a global pandemic. As the demand for poultry meat and products increases worldwide, commercial and backyard farms face the ongoing challenge of producing an essential supply of animal protein for human consumption. Current data suggest that the next influenza pandemic is likely to emerge from a novel viral strain to which the human population will have no pre-existing immunity. To reduce the transmission of this novel viral strain, continuous real-time monitoring efforts need to be integrated worldwide. With the advent of new technology, vaccine design can be refined using consensus-based algorithms, target conserved epitopes, and neutralizing epitopes-based approaches to produce better vaccines against viral pathogens with genetically diverse lineages. Consequently, more investment in avian influenza viruses is essential to develop vaccines and therapeutics and to enhance efficient preventive measures, such as tighter control of illegal poultry trafficking.

**Author Contributions:** Conceptualization, M.G. validation, F.S. and M.C. investigation, E.I., A.B., N.P. and L.B. resources, M.G.; data curation, M.C. writing—original draft preparation, E.I. and L.B.; writing—review and editing, M.G., E.I. and L.B.; supervision, M.G., M.C. and F.S. All authors have read and agreed to the published version of the manuscript.

**Funding:** This research received no external funding.

**Institutional Review Board Statement:** Not applicable.

**Informed Consent Statement:** Not applicable.

**Data Availability Statement:** Not applicable.

**Acknowledgments:** M.G. is funded by PON "Ricerca e Innovazione" 2014–2020.

**Conflicts of Interest:** The authors declare no conflict of interest.

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
