# Peer review of "Avian Influenza: Could the H5N1 Virus Be a Potential Next Threat?"

_2036-7481, doi:10.3390/microbiolres14020045_

Round 1

Reviewer 1 Report

In this review manuscript, Imperia et.al summarized the H5N1 epidemiology, cross-species transmission of H5N1, and current existing countermeasures in controlling influenza. Through these, they evoke for an urgent investment in genomic surveillance strategies to timely shape the emergence of any potential viral pathogen which in turn is the key to epidemic/pandemic preparedness.

Major comments:

Overall, the conclusion is not well supported by the evidence presented in the manuscript. The logistic of this manuscript is not easy to follow. The title of this manuscript is very attractive, but the content is not. It should be more focusing on the H5N1 based on the title.

Minor comments:

Sometimes “HPAIV” but sometimes “HPAI AIV”; Sometimes “AVI” but sometimes “AIV”;

Line 47: the number of cases presented here is not to cover the whole global but only the Western Pacific Region. Please revise all the statement you made in the text.

Line 50-52: These statements are very confusing. Please revise carefully.

Line 53-54: There are 4 types of influenza viruses now: A, B, C, and D. Influenza D virus was first identified in pigs and then found prevalent in cattle. 

Line 127-128: should be “reverse transcription-polymerase chain reaction” and abbreviate as “RT-PCR”.

Line 83-130: The logistics of this section is very hard to follow. Please revise carefully.

Line 140-141: The Antigenic and Genetic Characterization should have done for the Swine-Origin 2009 A(H1N1)Influenza Viruses.

Line 148-158: This paragraph must be mislocated here. It should be in the introduction.

Line 131: The subtitle of this section is “Avian Influenza Virus: evolution and genome”. The genome of the influenza A and B virus were mentioned very briefly and then the function of each viral protein was described in detail. There is not much of “evolution” that I can read in this section. And this section seems can be merged into the introduction.

Line 208: “1)” is typo?

Line 211: The direct transmission of AIVs to human from what?

Line 216: “avian cells”, is this referring to any avian cell line? Avian receptors are also found in human respiratory tract. It is generally described as avian receptors and human receptors but not as avian cells or human cells.

Line 235-238: Which virus was used?

Line 195: “Virologic characteristics of human H5N1”, this section mainly talked about the transmission of H5N1 virus to human and the molecule signature contributing the transmission. What are the virologic characteristics referring to?

Line 258-261: Fig. 2B which is an influenza virus life cycle, seems not necessary. 

There is too much information about the vaccine in the section 4 but not much related to the vaccine against H5N1.

The logistics should be checked throughout the manuscript.

Author Response

Response to Reviewer 1 Comments

Thank you for suggestions, we used the “Track Changes” function in Microsoft Word to highlight our modifications in the full text.

Major Comments:

Sometimes “HPAIV” but sometimes “HPAI AIV”; Sometimes “AVI” but sometimes “AIV”;

Reply: We thank the reviewer for his/her attention to the text and content. We modify the text using a unique acronym for Avian influenza virus (AIV) and for high pathogenic avian influenza virus (HPAIV).

Line 47: the number of cases presented here is not to cover the whole global but only the Western Pacific Region. Please revise all the statement you made in the text.

Reply: We appreciate the reviewer comment. We have revised all statement as suggested (from line 47 to 49).

Line 50-52: These statements are very confusing. Please revise carefully.

Reply: We thank the reviewer for this comment. We have revised it as suggested (lines 50 to 51).

Line 53-54: There are 4 types of influenza viruses now: A, B, C, and D. Influenza D virus was first identified in pigs and then found prevalent in cattle.

Reply: We thank the reviewer for this comment. We have added influenza D and a brief description, as suggested.

Line 127-128: should be “reverse transcription-polymerase chain reaction” and abbreviate as “RT-PCR”.

Reply: We thank the reviewer for this comment. According to your suggestion, we have added the acronym RT-PCR right after reverse transcription-polymerase chain reaction.

Line 83-130: The logistics of this section is very hard to follow. Please revise carefully.

Reply: We thank the reviewer for this comment. We have made substantial changes in this section.

Line 140-141: The Antigenic and Genetic Characterization should have done for the Swine-Origin 2009 A(H1N1) Influenza Viruses.

Reply: We thank the reviewer for this comment. We have modified it as suggested.

Line 148-158: This paragraph must be mislocated here. It should be in the introduction.

Reply: We thank the reviewer for this comment. We have located the paragraph in the introduction, as suggested.

Line 131: The subtitle of this section is “Avian Influenza Virus: evolution and genome”. The genome of the influenza A and B virus were mentioned very briefly and then the function of each viral protein was described in detail. There is not much of “evolution” that I can read in this section. And this section seems can be merged into the introduction.

Reply: We thank the reviewer for this comment. We have modified the subtitle as suggested.

Line 208: “1)” is typo?

Reply: We thank the reviewer for this comment. It was a typo; we have now deleted it.

Line 211: The direct transmission of AIVs to human from what?

 Reply: We thank the reviewer for this comment. We have added additional details.

Line 216: “avian cells”, is this referring to any avian cell line? Avian receptors are also found in human respiratory tract. It is generally described as avian receptors and human receptors but not as avian cells or human cells.

 Reply: We thank the reviewer for this comment. We have made the necessary changes.

Line 235-238: Which virus was used?

Reply: We thank the reviewer for bringing this up. Modified as required.

Line 195: “Virologic characteristics of human H5N1”, this section mainly talked about the transmission of H5N1 virus to human and the molecule signature contributing the transmission. What are the virologic characteristics referring to?

Reply: We thank the reviewer for this comment. We have modified the title of this section.

Line 258-261: Fig. 2B which is an influenza virus life cycle, seems not necessary. 

Reply: We thank the reviewer for this comment. We have modified it as suggested.

There is too much information about the vaccine in the section 4 but not much related to the vaccine against H5N1.

Reply: We appreciate the reviewer comment and we have made substantial changes in this section.

Reviewer 2 Report

A very interesting and relevant theme!

But:

The Review is very small. 

Thy name is "H5N1 avian influenza", but there is a lot of well-known information about all types of influenza in the text.

It is very difficult to trace the authors' thoughts in the body of the article, which is indicated in the Conclusion.

A lot of interesting data, for example: lines 22-24 and lines 47-49, again lines 98-100, again lines 106-108, again lines 109-111. 

This is not an analysis of the literature on influenza, but in more cases bibliographic description ( Chapter 2 especially).

For what purpose is the virus genome described (Chapter 3, starting from line 132)? This is not discussed further anywhere.

Two chapters 3.

All chapters are just a description, without a purpose. The impact of the introduction of a global genomic surveillance system that allows timely identification and characterization of new options that will guide the global response to the epidemic is not presented.

Author Response

Response to Reviewer 2 Comments

 Thank you for suggestions and recommendations, we used the “Track Changes” function in Microsoft Word to highlight our modifications in the full text.

The Review is very small. 

Reply: We thank the reviewer for this comment. The review falls within the characteristics (including the word counts) required by the journal guidelines.

Thy name is "H5N1 avian influenza", but there is a lot of well-known information about all types of influenza in the text.

Reply: We thank the reviewer for this comment.

It is very difficult to trace the authors' thoughts in the body of the article, which is indicated in the Conclusion.

Reply: We appreciate the reviewer comment and we have made extensive changes in the whole text to make it more cohesive.  

A lot of interesting data, for example: lines 22-24 and lines 47-49, again lines 98-100, again lines 106-108, again lines 109-111. 

Reply: We appreciate the reviewer's comment and have carefully revised the entire manuscript to eliminate any potential duplicate sections and to enhance the cohesiveness of the text.

This is not an analysis of the literature on influenza, but in more cases bibliographic description (Chapter 2 especially).

Reply: We appreciate the reviewer comment and we made sustancial changes to the entire manuscript.

For what purpose is the virus genome described (Chapter 3, starting from line 132)? This is not discussed further anywhere.

Reply: We appreciate the reviewer's comment. In Chapter three, we discuss genomic epidemiology, and we believe it is crucial to describe the genome structure in order to explain how the accumulation of point mutations, as well as reassortment and recombination events, can lead to the emergence of novel strains.

Two chapters 3.

Reply: We thank the reviewer for this comment. We have modified the numbers of chapters as suggested.

All chapters are just a description, without a purpose. The impact of the introduction of a global genomic surveillance system that allows timely identification and characterization of new options that will guide the global response to the epidemic is not presented.

Reply: We thank the reviewer for this comment. We have made substantial alteration to all the sections with the aim of improving our final draft.

Round 2

Reviewer 1 Report

I don't have more comments for authors.

I suggest authors to check the words and sentences:

For example: 

Line 163: "China's Hong Kong"? 

Line 163: "H5N1 was first isolated", I understand that authors refer to H5N1 influenza virus, I feel something is missing when read the sentence.

Author Response

Reviewer 1:

I suggest authors to check the words and sentences: For example: 

Comment 1: Line 163: "China's Hong Kong"? 

Reply: Thank you for your review. We have modified the sentence to read: “In 1997, H5N1 was initially discovered in a three-year-old boy from China, who suffered from acute pneumonia and respiratory distress syndrome. Following the first diagnosis, an additional seventeen patients were identified, of which six fatalities were reported.”

Comment 2: Line 163: "H5N1 was first isolated", I understand that authors refer to H5N1 influenza virus, I feel something is missing when read the sentence.

Reply: We thank the reviewer to bring this up. As stated above we have now made the necessary changes to this sentence.

Reviewer 2 Report

Remarks:

Replace the information in the Abstract (line 22-24) about the number of cases of the disease (should be the same with line 61-63).

Part 2. Need started line 186 "By the end of 2006" or similar. Here and next the chronology of data presentation is unclear. Specify the period (calendar data) on which the data is presented!

Line 210: Specify the location (country and city) of the CRN.

Part 5. It is necessary to structure the text – discuss and describe vaccines for humans and separately vaccines for animals (including birds).

I think that at the end of each part you need to briefly write a conclusion. This will explain the prerequisites for the "Conclusion". Now the Parts have no connection and the need for their presence is unclear.

Author Response

Reviewer 2:

Comment 1: Replace the information in the Abstract (line 22-24) about the number of cases of the disease (should be the same with line 61-63).

Reply: Done.

Comment 2: Part 2. Need started line 186 "By the end of 2006" or similar. Here and next the chronology of data presentation is unclear. Specify the period (calendar data) on which the data is presented!

Reply: We appreciate the reviewer comment, and we apologize the lack of clarity. We have r           revised the text to make it more cohesive which now reads:

“In late 2003 and early 2004, an H5N1 outbreak occurred in poultry farms across several countries, including Cambodia, China, Indonesia, Japan, Laos, South Korea, Thailand, and Vietnam, resulting in two deaths among two confirmed and one probable case [21, 22, 23, 24]. The virus subsequently spread to ten other Asian countries in 2004 due to the poultry trade. In Thailand, 17 cases and 12 deaths were reported, while in Vietnam, there were 28 cases and 20 deaths [25]. The virus then spread further afield, reaching Central Asia, South Asia, the Middle East, and parts of Africa in 2005 via migratory birds [26].

As of November 2003, a total of 861 human cases of H5N1 infection and 455 deaths have been reported from 17 countries, with a cumulative mortality rate of more than 50% [27].

The first H5N1 case in Egypt was confirmed in March 2006, with the Egyptian Ministry of Health reporting 6,355 suspected cases of H5N1 infection and 112 confirmed cases resulting in 36 deaths between March 2006 and March 2009, according to a WHO report. Of the confirmed cases, all except for two were linked to infected poultry [28].”

Comment 3: Line 210: Specify the location (country and city) of the CRN.

Reply: Done.

Comment 4: Part 5. It is necessary to structure the text – discuss and describe vaccines for humans and separately vaccines for animals (including birds).

I think that at the end of each part you need to briefly write a conclusion. This will explain the prerequisites for the "Conclusion". Now the Parts have no connection and the need for their presence is unclear.

Reply: We appreciate the reviewer's feedback, but respectfully disagree. In response to the first round of revisions, we made extensive changes to the entire text and had a native English-speaking scientist proofread it thoroughly. We also made additional revisions to enhance the coherence of all sections.